# Gandouling Mitigates CuSO_4_-Induced Heart Injury in Rats

**DOI:** 10.3390/ani12192703

**Published:** 2022-10-08

**Authors:** Shuzhen Fang, Wenming Yang, Kangyi Zhang, Chuanyi Peng

**Affiliations:** 1University Hospital, Anhui Agricultural University, 130 Changjiang Road West, Shushan District, Hefei 230036, China; fangshuzhen@ahau.edu.cn; 2Department of Neurology, The First Affiliated Hospital of Anhui University of Chinese Medicine, 117 Meishan Road, Shushan District, Hefei 230031, China; 3School of Tea and Food Science & Technology, Anhui Agricultural University, 130 Changjiang Road West, Shushan District, Hefei 230036, China; zhky202204@163.com (K.Z.); pcy0917@ahau.edu.cn (C.P.)

**Keywords:** Gandouling, copper-induced, heart injury, hepatolenticular degeneration, cardioprotective protection

## Abstract

**Simple Summary:**

Wilson’s disease (WD) is a rare autosomal recessive inherited disorder of copper (Cu) metabolism, which is one of the few neurogenetic diseases with a cure. Cu deposition in the heart tissue is a mechanism of cardiac involvement. Importantly, Gandouling (GDL) effectively promotes the excretion of intracellular Cu. Therefore, the purpose of this study was to evaluate the protective effects of GDL on heart injuries and the underlying mechanisms in a copper sulfate (CuSO4)-induced animal model. The results showed that the protective effect of GDL on the heart was superior to that of penicillamine.

**Abstract:**

We assessed the protective effects of Gandouling (GDL) on copper sulfate (CuSO_4_)-induced heart injuries in Sprague–Dawley rats, which were randomly divided into the control, CuSO_4_, GDL + CuSO_4_ and penicillamine + CuSO_4_ groups. The rats received intragastric GDL (400 mg/kg body weight) once per day for 42 consecutive days after 56 days of CuSO_4_ exposure, and penicillamine was used as a positive control. The levels of plasma inflammatory cytokines (IMA, hFABP, cTn-I and BNP) were determined using the enzyme-linked immunosorbent assay. The histopathological symptoms were evaluated using hematoxylin and eosin staining and transmission electron microscopy. To determine the underlying mechanism, Western blotting was conducted for the detection of the heme oxygenase 1 (HO-1) expression. The results revealed that GDL supplementation alleviated the histopathological symptoms of the rat heart tissue, promoted Cu excretion to attenuate impairment, and significantly decreased inflammatory cytokine levels in the plasma (*p* < 0.01). In addition, GDL increased the HO-1 expression in the rat hepatic tissue. The protective effect of GDL on the heart was superior to that of penicillamine. Overall, these findings indicate that GDL alleviates hepatic heart injury after a Cu overaccumulation challenge, and GDL supplements can be beneficial for patients with Wilson’s disease.

## 1. Introduction 

Hepatolenticular degeneration or Wilson’s disease (WD) is a rare autosomal recessive inherited disorder of copper (Cu) metabolism [1,2]. It was first defined by Kinnear Wilson in 1912 [3]. WD, characterized by the overaccumulation of Cu in the kidneys, brain, liver, and other vital organs, results from a ceruloplasmin synthesis obstacle and abnormal biliary Cu excretion induced by the *ATP7B* mutation [1,4]. The morbidity rate of WD is approximately 1–3 per 30,000 individuals worldwide [5,6]. Because of the latent onset of WD, early diagnosis and treatment can delay or stop disease progression. 

Studies have mainly focused on the effects of WD on the liver, kidneys, brain, skeleton, cornea, and nervous system [1,2,7,8,9], with few studies evaluating the concurrent symptoms of cardiac injuries. Damages caused by WD can be categorized into cardiovascular disorders, such as cardiomyopathy, arrhythmia, and sudden cardiac death, and autonomic nervous system disorders. Therefore, in addition to the brain and liver, the heart is one of the primarily affected organs in patients with WD [10]. Azevedo evaluated the heart biopsy of a patient with WD and mild liver disease and discovered moderate myocardial injury. Moreover, the Cu content in the patient’s heart muscles was 10 times that in normal heart muscles [11], suggesting that the myocardial injury was related to Cu deposition.

WD is one of few neurogenetic diseases with a cure. Regarding conventional Western medicine, metal chelation therapy, such as penicillamine therapy [9,12,13], is the first line of treatment for reducing Cu accumulation. However, this therapy causes allergic reactions and immunologic injuries [9,13], thereby limiting its clinical application. Traditional Chinese herbal medicines, such as Xiaoyao powder, Bushen Jianpi decoction, and Gandou [2]), are advantageous because of their multi-channel, multi-target, and synergistic effects on WD [14,15,16]. Gandouling (GDL), a Chinese herbal medicine, was developed by the First Affiliated Hospital of Anhui University of Chinese Medicine [17] and has been widely used in the treatment of WD for decades with great therapeutic effect and less side effects for the past many years. In our previous studies, GDL improved cerebrovascular [9] and hepatic injuries [1], promoted the proliferation and differentiation of neural stem cells [2], and inhibited excessive mitophagy [18]. This study evaluated the protective effects of GDL on heart injuries and the underlying mechanisms in a copper sulfate (CuSO_4_)-induced animal model.

## 2. Materials and Methods

### 2.1. Chemical Reagents

This study used GDL tablets (0.3 g/tablet) produced by the First Affiliated Hospital of Anhui University of Chinese Medicine, as described previously [2]. Penicillamine (0.125 g/tablet) was purchased from the Xinyi general pharmaceutical factory of Shanghai Fosun Pharmaceutical (Group) Co., Ltd (Shanghai, China). CuSO_4_ was of the highest grade and was purchased from Sigma (St. Louis, MO, USA). Heme oxygenase 1 (HO-1) was purchased from Shanghai Yuanye Biotechnology Co., Ltd. (Shanghai, China). Ischemia-modified albumin (IMA), cardiac troponin I (cTn-I), and heart-type fatty acid binding-protein (hFABP) were obtained from Nanjing Jiancheng Bioengineering Institute (Nanjing, China). Natriuretic peptide (BNP) commercial kits were purchased from Shanghai Yuanye Biotechnology Co., Ltd (Shanghai, China). Other reagents used were of the highest available grade.

### 2.2. Animals and Treatments

A total of 50 healthy male Sprague–Dawley rats weighing 200 to 260 g were obtained from the Experimental Animal Center of Anhui Medical University (Hefei, China). The animals were provided food and water ad libitum and were housed in a controlled environment at 22 ± 1 °C with 50–60% humidity and under a 12 h light–dark cycle (8:00–20:00 light conditions). All animal experiments were performed in accordance with the National Institutes of Health Guide for the Care and Use of Laboratory Animals (NIH Publications No. 8023, revised 1978) and were reviewed and approved by the Institutional Animal Care and Use Committee of Anhui University of Chinese Medicine (ethical approval code: AUCM2014030311).

Experiments were designed to investigate the cardioprotective effects of GDL on CuSO_4_-induced subacute heart injury (Figure 1). The dosages of GDL and penicillamine were used as previously described [9,18]. After one week of acclimation on laboratory food, the rats were randomly divided into four groups (*n* = 10 per group): (a) control (saline only), (b) CuSO_4_, (c) GDL (intragastrically administered, 400 mg/kg body weight [b.w.]) + CuSO_4_, and (d) penicillamine (intragastrically administered, 100 mg/kg b.w.) + CuSO_4_. Group (a) served as a control and had free access to saline solution. Animals in the CuSO_4_-exposed groups, including the groups (b), (c), and (d), received 0.185% (*m*/*v*) CuSO_4_ aqueous solution as drinking water and 1 g/kg (*m*/*m*) CuSO_4_ in the diet for 56 consecutive days before GDL and penicillamine exposure. CuSO_4_ exposed through drinking water and diet continued in the subsequent 42 days.

The rats were anesthetized and sacrificed by cervical dislocation 24 h after the end of the stimulation period. Plasma was obtained through the centrifugation of blood samples at 3000 rpm for 10 min and was stored at −80 °C until analysis [19]. Some heart tissues were rinsed in ice-cold phosphate-buffered saline and were stored at −80 °C for Cu analysis, and the remaining tissues were fixed with 4% paraformaldehyde and 2.5% glutaraldehyde at 4 °C for histopathological analyses.

### 2.3. Measurements for Assessingheart Damage

Blood samples were collected from the abdominal aorta. The levels of inflammatory cytokines, including IMA, hFABP, cTn-I, and BNP, in the plasma supernatant were determined using the enzyme-linked immunosorbent assay (ELISA) according to the manufacturer’s protocols.

### 2.4. Determination of Cu Concentration

The wet decomposition method with HNO_3_,H_2_SO_4_, and H_2_O_2_ was employed to digest the heart samples [20]. The digestive solutions were filtered through 0.45 μm cellulose acetate membranes before being injected into an apparatus. Cu concentrations were determined using an inductively coupled plasma-optical emission spectrometer (ICP-OES, iCAP 6300 Series, Thermo, Waltham, MA, USA) [20], and the standard curve method was used for quantification. Each sample was analyzed in triplicate.

### 2.5. Histopathological Examination of the Heart Tissues 

Heart histology was assessed using hematoxylin and eosin (H&E) staining (Wang et al., 2020). Paraffin blocks of the heart tissue were cut into 4 μm thick sections (LKB-2088, Bromma, Sweden), and stained with H&E according to the standard protocol. The pathological analysis of the heart was performed as described in a study [19,21].

### 2.6. Ultrastructural Examination of the Hearttissue by Using a Transmission Electron Microscope

Samples were prepared as described previously [22] with some minor revisions. In brief, the heart tissues were fixed in 2.5% precooled glutaraldehyde with 0.1 mol/L cacodylate buffer (pH = 7.4), followed by post-fixation in 1% osmium tetroxide and dehydration. Subsequently, ultrathin sections (60–80 nm) were randomly prepared on Cu grids, stained with lead citrate and uranyl acetate, and observed using a transmission electron microscope (JEM-1230, Jeol, Tokyo, Japan).

### 2.7. Western Blot Analysis 

Proteins were extracted from the frozen heart tissue lysates by using radioimmunoprecipitation assay buffer (Sigma-Aldrich, Shanghai, China) containing 1% phosphatase inhibitor cocktail (Sigma-Aldrich, Shanghai, China) and protease (Sigma-Aldrich, Shanghai, China). The lysates were centrifuged at 13,000× *g* for 20 min at 4 °C [19], and the proteins were separated using an SDS-PAGE Gel Preparation Kit (Sangon Biotech, Shanghai, China) and were transferred to PVDF membranes. The membranes were incubated overnight with primary HO-1 or β-actin monoclonal antibodies (Proteintech, Wuhan, China) at 4 °C, followed by incubation with a secondary antibody (Proteintech, Wuhan, China). The target antigens were detected using standard chemiluminescence methods, and the band intensities were measured using Image J software (NIH Shareware, Ver. 1.47t, Bethesda, MD, USA) [18,19]. The relative intensity of the target band was defined as the ratio of the target band intensity to β-actin (internal control) band intensity.

### 2.8. Statistical Analysis

Data from the independent experiments are presented as means ± standard deviations (SDs), and graphs were constructed using Prism 5.0 (GraphPad Software, La Jolla, CA, USA). On the basis of the results of Bartlett’s test for equal variance, one-way analysis of variance was performed for determining differences among multiple groups. A post hoc analysis was conducted using either Dunnett’s or Tukey’s test for multiple comparisons. All statistical analyses were conducted using the GraphPad software, and two-sided *p* values of <0.05 and <0.01 were considered statistically significant [19,21].

## 3. Results

### 3.1. GDL Protect against CuSO_4_-Induced Heart Injury in Rats

To determine the effects of GDL on a CuSO_4_-induced heart injury, GDL was intragastrically administered to the rats. The plasma levels of plasma IMA, hFABP, cTn-I, and BNP were determined using ELISA after 42 days of exposure. The plasma levels of IMA, hFABP, cTn-I, and BNP significantly increased after the CuSO_4_ exposure (*p* < 0.01). GDL significantly attenuated the increases (*p* < 0.01; Figure 2A–D) and exerted a much stronger effect than the positive control, that is, penicillamine (*p* < 0.01). The results indicated that GDL provided more powerful protection against a CuSO_4_-induced heart injury. To confirm the protective effects of GDL on a CuSO_4_-induced heart injury, we analyzed the histopathological changes in the rat heart with H&E staining (Figure 3). Compared with the control heart, the heart in the CuSO_4_-induced heart injury group exhibited some characteristics, such as slightly edematous, lightly stained, and even disappearing cells (Figure 3C,D). Both GDL and penicillamine significantly eliminated these pathological alterations. The histopathological observations in the GDL-treated rats were consistent with the levels of plasma inflammatory cytokines.

### 3.2. Effect of GDL on Ultrastructural Changes in the Heart Tissue

To further explore the protection against a CuSO_4_-exposed heart injury from the GDL protective effects, the ultrastructure of myocardial tissue was observed and examined. As can be seen from Figure 4, compared with the control (Figure 4A,B), Cu overaccumulation led to a gradual rupture of the myofilament in the myocardial tissue, as well as mitochondrial vacuolation and obscurity to disappearance of the ridge line (Figure 4C,D). Interestingly, GDL stimulation significantly eliminated these pathological alterations (Figure 4E,F), which makes the protection effect better than the penicillamine stimulation, as shown in Figure 4G,H.

### 3.3. GDL Promotes Cu Excretion in the Heart Tissue to Attenuate Impairment 

Cu overaccumulation in the heart tissues may contribute to WD development [3,4]. To determine the effects of GDL on Cu excretion in the heart tissue, Cu levels in the heart tissues were determined after 42 h of exposure by using an ICP-OES. The Cu level in the CuSO_4_-induced heart tissues was 8.28 ug/g, which was higher than that in the control heart tissue (6.01 ug/g, *p* < 0.01); however, GDL significantly promoted Cu excretion to achieve the level of 6.60 ug/g (*p* < 0.01; Figure 5). The ability of GDL against Cu accumulation was equivalent to that of penicillamine (*p* > 0.05).

### 3.4. GDL Augments HO-1 Expression in the Heart Tissue 

As shown in Figure 6, CuSO_4_ exposure significantly downregulated the expression of HO-1, which ranged from 1.35 to 0.58 (*p* < 0.01). However, GDL augmented the HO-1 expression, with the effect being significantly higher and better than that of penicillamine. 

## 4. Discussion

GDL, a traditional Chinese herbal medicine, was developed by the First Affiliated Hospital of Anhui University of Chinese Medicine and approved by the National Medical Product Administration (approval number: Z20050071). GDL was commercialized and reported to exhibit several biological functions, such as antioxidant, blood activation, anti-inflammatory, and neuroprotective functions and promotion of Cu excretion [1,2,9,14]. However, few studies have evaluated the cardioprotective effects of GDL, and heart injury is a complication of Cu overaccumulation. Thus, this work aimed to assess the protective effects on heart injury from GDL and underlying mechanisms in an animal model.

We investigated the protective effects of GDL on a CuSO_4_-induced heart injury by evaluating the histopathological changes, ultrastructural changes, plasma inflammatory responses, and HO-1 expression changes. Cu deposition in the heart tissue is a mechanism of cardiac involvement in patients with WD. It is reported that Cu overaccumulation can damage the organs [1,2,7,8,9].

GDL consists of *Radix et Rhizoma Rhei*, *Rhizoma coptidis*, *Radix Scutellariae*, *Salvia*
*miltiorrhiza*, *Curcuma longa*, *Curcuma aromatica*, and *Caulis Spatholobi*. GDL effectively promotes the excretion of intracellular Cu [14]. Modern medical studies have demonstrated that *Radix et Rhizoma Rhei* and *Rhizoma coptidis* can promote bile secretion and excretion, thereby facilitating the excretion of Cu in bile acids through feces. In addition, berberine in *Rhizoma coptidis* improves oxygenation and reduces oxygen consumption in heart tissues [23], as well as *Salvia miltiorrhiza* [24,25]. *Curcuma longa* and *Curcuma aromatica* inhibit myocardial fibrosis, inflammatory reaction and apoptosis to achieve myocardial protection [26,27,28]. Through ingredient-target network analysis, Zhang et al. identified 324 active compounds in the GDL tables used for WD treatment [29]. GDL has abundant physiologically active components to protect the heart.

Cu is one of the essential trace elements in the human body, and Cu deficiency may lead to heart issues, such as cardiac hypertrophy and compromised contractile function [30,31,32]. Cu overaccumulation causes irreparable damage to several organs and systems of the human body, and the heart exhibits high sensitivity to Cu [33]. CuSO_4_ exposure significantly increased the plasma levels of inflammatory cytokines, including IMA, hFABP, cTn-I, and BNP, which are strongly associated with increased risks of myocardial injury and ischemia. It is well known that IMA has been evaluated as a potential biomarker of early myocardial ischemia in clinical models of heart injury recognized by the United States Food and Drug Administration [34,35]; hFABP is a specific biomarker and immediately functions as a myocardial fatty acid transporter when myocardial injury occurs [36]; cTn-I, a specific contractile protein on the myocardial fibers, is a non-enzymatic serum marker for detecting myocardial injury with high specificity and sensitivity [37]; and BNP is a validated risk marker in cardiac diseases and can provide a remarkable guidance in cardiovascular complications prediction [38]; HO-1, a crucial part of the Nrf2/HO-1 signaling pathway, plays a physiological role through its anti-inflammatory and anti-oxidant actions [19,39]. In previous studies, HO-1 up-regulation decreased myocardial ischemia-reperfusion injury and effectively promoted the recovery of cardiac function through the activation of the Nrf2/HO-1 pathway [40,41]. Studies have revealed that GDL promoted neurogenesis and alleviated symptoms by the activation of the Nrf2/HO-1 pathway in WD models [2]. In the current study, GDL effectively promoted Cu excretion, reducing the Cu concentration in the heart tissue and inflammatory responses in the blood to achieve cardioprotective effects. Future studies should analyze more indexes of the Nrf2/HO-1 pathway to corroborate our initial findings.

## 5. Conclusions

The protective effects of GDL against CuSO_4_ exposure complication, i.e., heart injury, were investigated. GDL supplementation ameliorated the histopathological symptoms of the heart tissue, promoted Cu excretion, and decreased inflammatory cytokines levels in the plasma significantly. For a practical application of the results obtained herein, GDL dramatically alleviates hepatic heart injury, with a protective effect even better than that of penicillamine, caused by Cu overaccumulation in the heart tissue.

## Figures and Tables

**Figure 1 animals-12-02703-f001:**
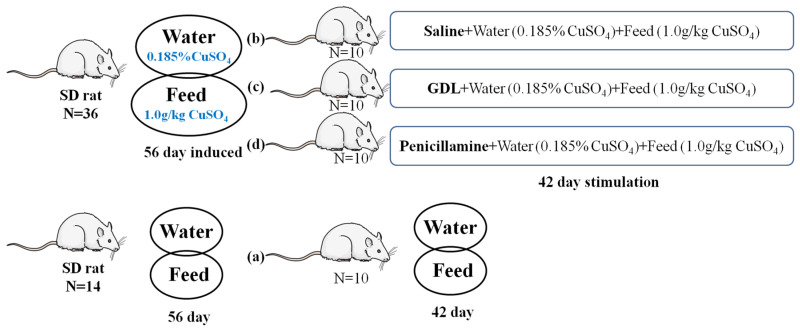
Schematic of the experimental design.

**Figure 2 animals-12-02703-f002:**
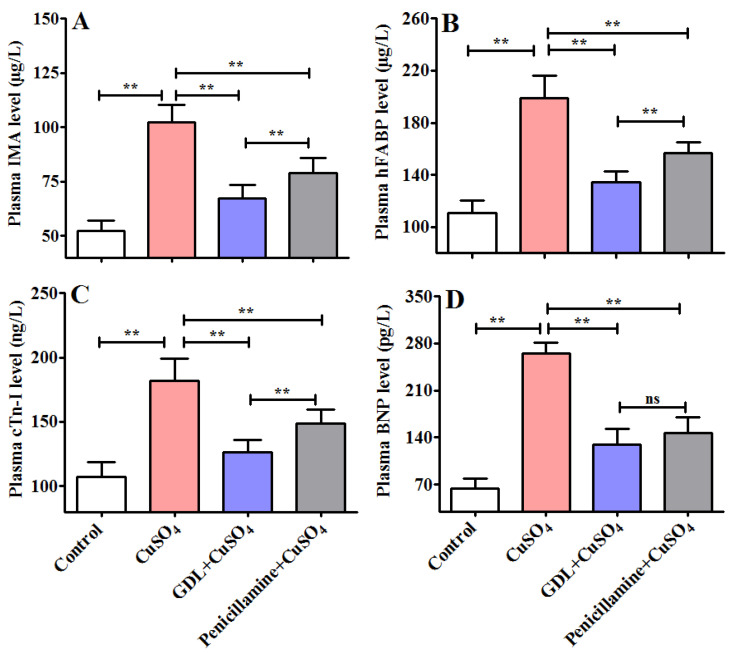
Effects of Gandouling (GDL) on copper sulfate (CuSO_4_) -induced heart injury in rats. (**A**) Plasma ischemia-modified albumin (IMA). (**B**) Plasma heart-type fatty acid binding-protein (Hfabp). (**C**) Plasma cardiac troponin I (cTn-I). (**D**) Plasma Natriuretic peptide (BNP). (**A**–**D**) Data are presented as mean ± SD (*n* = 10). ** *p* < 0.01. ns: no significance.

**Figure 3 animals-12-02703-f003:**
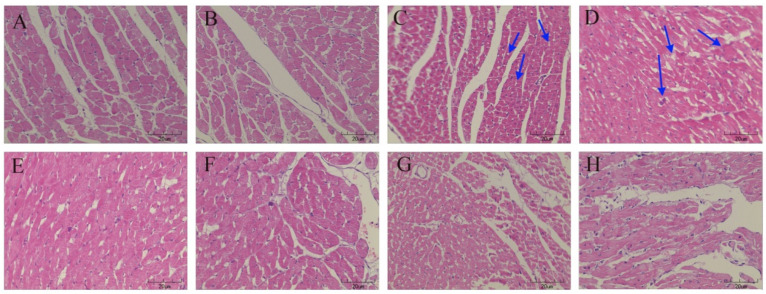
The results of pathological changes in the rat heart tissue after different treatments. (H&E staining × 400). (**A**,**B**) Control; (**C**,**D**) CuSO_4_; (**E**,**F**) GDL + CuSO_4_; (**G**,**H**) Penicillamine + CuSO_4_. The arrows indicated the pathological changes in the rat heart tissue after CuSO_4_ exposure.

**Figure 4 animals-12-02703-f004:**
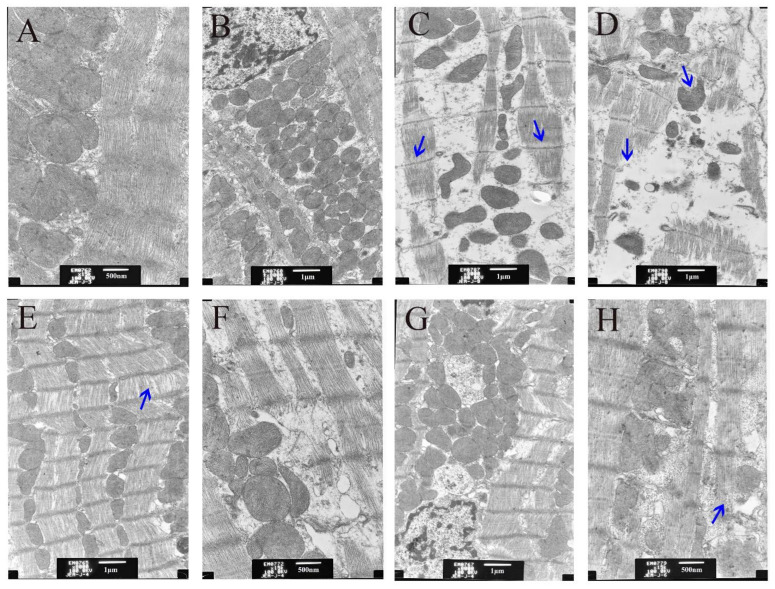
Ultrastructure of the rat heart tissue after different treatments under electron microscope. (**A**,**B**) Control; (**C**,**D**) CuSO_4_; (**E**,**F**) GDL + CuSO_4_; (**G**,**H**) Penicillamine + CuSO_4_. The arrows indicated the ultrastructural changes after CuSO_4_ exposure, GDL and penicillamine treatments.

**Figure 5 animals-12-02703-f005:**
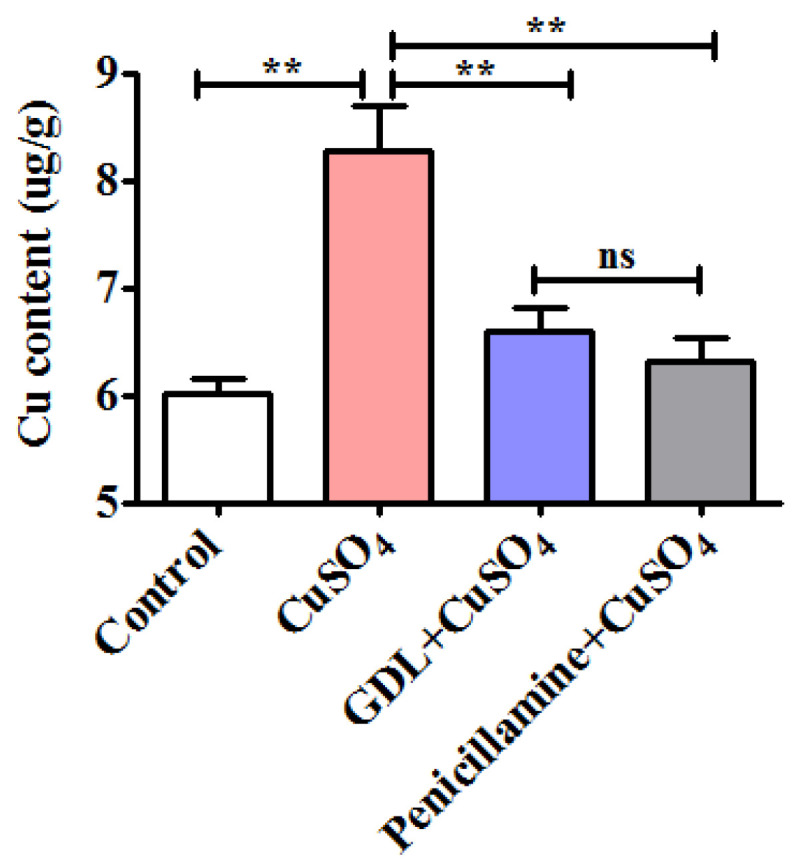
Effects of GDL on Cu excretion in the heart tissue. Data are presented as mean ± SEM (*n* = 10). ** *p* < 0.01. ns: no significance.

**Figure 6 animals-12-02703-f006:**
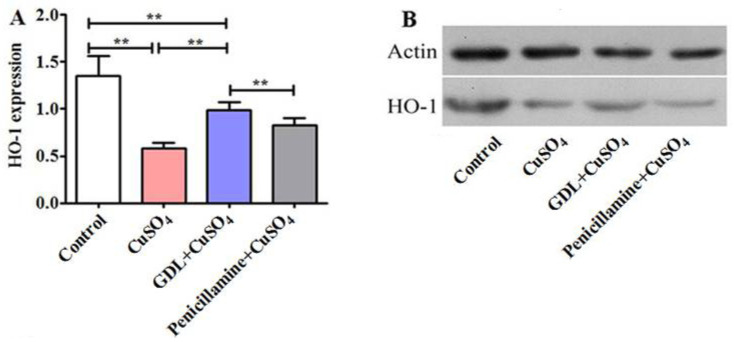
Effect of HO-1 protein in the heart tissue after different treatments (**A**). HO-1 and Action (**B**). Data are presented as mean ± SEM (*n* = 10). ** *p* < 0.01.

## Data Availability

Not applicable.

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
