# Peer review of "Gandouling Mitigates CuSO4-Induced Heart Injury in Rats"

_animals, 2022, doi:10.3390/ani12192703_

Round 1
Reviewer 1 Report
The manuscript is so interesting and presents a good idea. It provides another chleating agent against cupper beside penicillamine.
I hope Gandouling will be available and its uses will be implemented.
Author Response
Q1: The manuscript is so interesting and presents a good idea. It provides another chleating agent against cupper beside penicillamine.
I hope Gandouling will be available and its uses will be implemented.
Response: Thanks very much for your recognition and encouragement of our research work, giving us a lot of motivation. We will continue to focus on the Health function of Gandouling.

Reviewer 2 Report
This study entitled “Gandouling Tablets Mitigate the Copper-Overloaded-Induced Heart Injury in Rat”. The authors found that Gandouling (GDL) supplementation alleviated the histopathological symptoms of myocardial tissue and promoted Cu excretion to attenuate impairment in CuSO4-in-duced rat, decreasing inflammatory cytokines levels in the plasma significantly.
Some comments as below:
1. Only representative figures of pathological changes were found in Figure 3 and 4. Suggest analyze and compare the pathological score between different groups.
2. Section 3.3 described “the ability to repel Cu of GDL was even more 196 perfect than penieillamine”. However, it is not comparable to the result of Fig. 5, no difference between GDL and penicillamine on Cu excretion in myocardial tissue.
3. What is the symbol of arrows in figure 3 and 4 for? Please describe it in the figure legends.
4. In the Figure 6 legend, there is no mean ± SEM, animal numbers and p value. Please add them.
5. The English should be re-polished again, such as “these results show that GDL alleviates “hepatic” heart injury after Cu overaccumulation challenge… in the Abstract”.
Author Response
Q1: Only representative figures of pathological changes were found in Figure 3 and 4. Suggest analyze and compare the pathological score between different groups.
Responses: We are very grateful to you and your positive comments. As suggested, we revised in the revised manuscript.
Q2: Section 3.3 described “the ability to repel Cu of GDL was even more 196 perfect than penieillamine”. However, it is not comparable to the result of Fig. 5, no difference between GDL and penicillamine on Cu excretion in myocardial tissue.
Responses: We are very grateful to you and your positive comments. As suggested, we revised it into “The ability of GDL against Cu accumulation was equivalent to that of penicillamine (P > 0.05).” in the revised manuscript .
Q3: What is the symbol of arrows in figure 3 and 4 for? Please describe it in the figure legends.
Responses: Thanks very much for your great comments and suggestions. As suggested, we provided the annotations in the figure legends.
Q4: In the Figure 6 legend, there is no mean ± SEM, animal numbers and p value. Please add them.
Responses: Thanks very much for your great comments. As suggested, we provided mean ± SEM, animal numbers and p value in the revised manuscript.
Q5: The English should be re-polished again, such as “these results show that GDL alleviates “hepatic” heart injury after Cu overaccumulation challenge… in the Abstract”.
Responses: We are very grateful to you and your positive comments. As suggested, we re-write and ask Wallace Academic Editing for language service, please find the language service certificate.

Reviewer 3 Report
Reviewer
The manuscript by Shu-zhen FANG et al. has investigated Gandouling Tablets Mitigate the Copper-Overloaded-Induced Heart Injury in Rat. The results showed that Gandouling (GDL) supplementation alleviated the histopathological symptoms of myocardial tissue and promoted Cu excretion to attenuate impairment in CuSO4-induced rat, decreasing inflammatory cytokines levels in the plasma significantly (P < 0.01). Additionally, GDL increased HO-1 protein expression in hepatic tissue.
After an exhaustive revision, the manuscript is Reconsider after major revision. In general, the study is closely connected to the journal's objectives. The study is very interesting.
After an exhaustive revision, the manuscript is Reconsider after major revision.
In general, the study is closely connected to the journal's objectives. The study is very interesting.
The section result, some subsections need to an important improve.
The section discussion is an important problem, since is very poor, the lines correspond to introduction, a few on explication of results, a few on comparison with others studies, and a few on discussion of the results obtained with respect to other studies. The selection of bibliography is appropriate to the content of the manuscript.
After close evaluation of the paper I suggest revision according to the next points:
1. Should be corrected Figure 2. “Figure 2. Effects of GDL on CuSO4-induced heart injury in rat.
(A) Plasma IMA. (B) Plasma hFABP. (C) Plasma cTn-I. (D) Plasma BNP.
(A–D) Data are presented as mean ± SD (n = 10). *P < 0.05 and **P < 0.01 versus comparison group.”
2. Should be corrected Introduction section “ 1. Introduction”
3. “2.6. Ultrastructural examination in myocardial tissue by transmission electron microscope
Samples were prepared accroding to Xu et al. (Xu, et al., 2020) with some minor re-128 vison. Brief, myocardial tissues were fixed in 2.5 % precooled glutaraldehyde with 0.1 mol/L cacodylate buffer (pH=7.4), post-fixed in 1.0 % osmium tetroxide and dehydrated. Then, ultrathin sections (60-80 nm) were randomly prepared on Cu grids, stained with lead citrate and uranyl acetate, and then observed assisted with transmission electron microscope (JEM-1230, Janpan).”
Was copper content determined to show excessive copper accumulation Cu overaccumulation led to a gradual rupture of myofilament in the myocardial tissue (Fig. 4C-D)?
3. Should be corrected Result section “3. Results
“3.3. GDL augment HO-1expression of myocardial tissue”
“3.3. GDL promote Cu excretion in myocardial tissue to attenuate impairment”
4. Should be corrected Discussion section
5. Should be corrected Reference:
19. Loudianos, G., Zappu, A., Lepori, M. B., Incollu, S., Dessi, V., Mameli, E., Garrucciu, G., De Virgiliis, S., & Cao, A. Wilson's disease in two consecutive generations: the detection of three mutated alleles in the ATP7B gene in two Sardinian families. Digestive and Liver Disease. (2013) 45(4): 342-345. doi:10.1016/j.dld.2012.10.017
and others
Author Response
Q1: After an exhaustive revision, the manuscript is Reconsider after major revision. In general, the study is closely connected to the journal's objectives. The study is very interesting.
Responses: We are very grateful to you and your positive comments.
Q2: Should be corrected Figure 2. “Figure 2. Effects of GDL on CuSO4-induced heart injury in rat.(A) Plasma IMA. (B) Plasma hFABP. (C) Plasma cTn-I. (D) Plasma BNP. (A–D) Data are presented as mean ± SD (n = 10). *P < 0.05 and **P < 0.01 versus comparison group.”
Responses: We are very grateful to you and your positive comments. As suggested, we revsied the Figure 2 in the revised manuscript.
Q3: Should be corrected Introduction section “ 1. Introduction”
Responses: We are very grateful to you and your positive comments. As suggested, we revsied the Introduction section in the revised manuscript.
Q4: “2.6. Ultrastructural examination in myocardial tissue by transmission electron microscope. Samples were prepared accroding to Xu et al. (Xu, et al., 2020) with some minor re-128 vison. Brief, myocardial tissues were fixed in 2.5 % precooled glutaraldehyde with 0.1 mol/L cacodylate buffer (pH=7.4), post-fixed in 1.0 % osmium tetroxide and dehydrated. Then, ultrathin sections (60-80 nm) were randomly prepared on Cu grids, stained with lead citrate and uranyl acetate, and then observed assisted with transmission electron microscope (JEM-1230, Janpan).” Was copper content determined to show excessive copper accumulation Cu overaccumulation led to a gradual rupture of myofilament in the myocardial tissue (Fig. 4C-D)?
Responses: We are very grateful to you and your positive comments. Yes, Cu overaccumulation led to a gradual rupture of myofilament in the myocardial tissue , as well as mitochondrial vacuolation and obscurity to disappearance of ridge line (Fig. 4C-D).
Q5: Should be corrected Result section “3. Results: “3.3. GDL augment HO-1expression of myocardial tissue”, “3.3. GDL promote Cu excretion in myocardial tissue to attenuate impairment”
Responses: We are very grateful to you and your positive comments. As suggested, we revsied the Result section in the revised manuscript.
Q6: Should be corrected Discussion section
Responses: We are very grateful to you and your positive comments. As suggested, we revsied the Discussion section in the revised manuscript.
Q7: Should be corrected Reference: Loudianos, G., Zappu, A., Lepori, M. B., Incollu, S., Dessi, V., Mameli, E., Garrucciu, G., De Virgiliis, S., & Cao, A. Wilson's disease in two consecutive generations: the detection of three mutated alleles in the ATP7B gene in two Sardinian families. Digestive and Liver Disease. (2013) 45(4): 342-345. doi:10.1016/j.dld.2012.10.017, and others
Responses: Thanks very much for your great comments and suggestions. As suggested, we revised and checked all the references format according to the latest format requirements from author guideline.
Reviewer 4 Report
Reviewer comments and suggestions
The study aimed to investigate Gandouling (GDL) exerts that may have potential protective effects against CuSO4-induced complications in Sprague Dawley rats. For this authors, divided the rat into four groups: control, CuSO4, GDL + CuSO4 and penicillamine + CuSO4. The rat received intragastric GDL (400 mg/kg b.w.) once per day for 42 consecutive days after CuSO4 56-day stimulation, and penieillamine was carried out as a positive control. The animal inflammatory cytokines, histopathological symptoms, and transmission electron microscopy (TEM) observation, western blotting (WB) was the technique used in the study.
The result noted that the GDL supplementation alleviated the histopathological symptoms of myocardial tissue and promoted Cu excretion to decrease impairment in CuSO4-induced rats, decreasing inflammatory cytokines levels in the plasma significantly (P < 0.01). Finally, the result showed that GDL alleviates hepatic heart injury after Cu overaccumulation challenge, it could be used for benefitting people with WD disease.
Overall, the manuscript needs a thorough revision for publication. Few concerns are below to be incorporated in the revised version of the manuscript.
- Line 7 Please check the typo error, Line 33, 197 and the manuscript has many typo errors, I advise the authors to proofread again this manuscript by a native English speaker
- Line 15 Heosin should be eosin
- Please follow the mdpi guidelines while citing the references. The author needs to modify all the cited references in the text. It should be in number, not authors et al
- Line 56 these all composition has been utilized in Li 2013 paper? if not try to add up a suitable reference.
- Line 61 If the authors know these extracts, they must have explained the ingredients present in the GDL
- Figure 1 Please draw it in a better way, the representation was poor
- Line 111 please explore these cytokines, at least a few points needed to be mentioned for the importance of these cytokines
- Line 141-142 which types of antibodies are used in the western and explain the expression of which protein
- Comments of all figures “The legend expression was poor in the figure, please modify”
- For Figure 4 The arrow used in the figure was discussed in the legend
- Discussion first para “same information was needed here, first para need to add the novelty of the study”.
- Line 230-231 These points should be present in the introduction
- Line 243-246 These points were already discussed, better to discuss your result and compare it with the similar findings from other published studies
- Line 253-257 not convinced with the written part, it should be written professionally, lastly, they directly suggested that "in the current study, please make a constant balance between the lines and provided information
- The journal style should be used by MDPI author guidelines. Abbreviated. please check all the references once again
Author Response
Q1:Line 7 Please check the typo error, Line 33, 197 and the manuscript has many typo errors, I advise the authors to proofread again this manuscript by a native English speaker.
Responses: We are very grateful to you and your positive comments. We ask Wallace Academic Editing for language service, please find the language service certificate.
Q2:Line 15 Heosin should be eosin
Responses: We are very grateful to you and your positive comments. As suggested, we revised and checked the whole manuscript.
Q3:Please follow the mdpi guidelines while citing the references. The author needs to modify all the cited references in the text. It should be in number, not authors et al
Responses: Thanks very much for your great comments and suggestions. As suggested, we revised and checked all the references format according to the latest format requirements from author guideline.
Q4: Line 56 these all composition has been utilized in Li 2013 paper? if not try to add up a suitable reference.
Responses: Thanks very much for your great comments. As suggested, we checked and added up a new reference of our our previous research.
Q5:Line 61 If the authors know these extracts, they must have explained the ingredients present in the GDL.
Responses: We are very grateful to you and your positive comments. As described, GDL is a traditional Chinese herbal medicine, mainly consists of Radix et Rhizoma Rhei, Rhizoma coptidis, Radix Scutellariae, Salvia Miltiorrhiza, Curcuma longa, Curcuma aromatica and Caulis Spatholobi. We provided these information in the disscuss part, however, the detailed proportion of ingredients involves commercial secret, because GDL is a commercial medicine. And the preparation of GDL was introduced in our previous research.
Dong, T., Wu, M. C., Tang, L. L., Jiang, H. L., Zhou, P., Kuang, C. J., Tian, L. W., & Yang, W. M. (2021). GanDouLing promotes proliferation and differentiation of neural stem cells n he mouse model of Wilson's disease. Bioscience Reports, 41(1).
https://doi.org/10.1042/BSR20202717
Q6:Figure 1 Please draw it in a better way, the representation was poor.
Responses: We are very grateful to you and your positive comments. As suggested,
we revised and improved Figure 1 in the revised manuscript.
Q7:Line 111 please explore these cytokines, at least a few points needed to be mentioned for the importance of these cytokines
Responses: We are very grateful to you and your positive comments. As suggested, we revised and supplemented some related informations in the revised manuscript.
Q8:Line 141-142 which types of antibodies are used in the western and explain the expression of which protein
Responses: We are very grateful to you and your positive comments. As suggested, we revised and supplemented the informations in the revised manuscript.
Q9:Comments of all figures “The legend expression was poor in the figure, please modify”
Responses: We are very grateful to you and your positive comments. As suggested, we revised the legend expression in the revised manuscript.
Q10:For Figure 4 The arrow used in the figure was discussed in the legend
Responses: We are very grateful to you and your positive comments. Cu overaccumulation led to a gradual rupture of myofilament in the myocardial tissue (Fig. 4C-D), as well as mitochondrial vacuolation and obscurity to disappearance of ridge line (Fig. 4C-D). GDL stimulation significantly eliminated these pathological alterations, which makes the protection effect better than penieillamine stimulation (Fig. 4G-H). As suggested, we revised some discussions in the revised manuscript.
Q11:Discussion first para “same information was needed here, first para need to add the novelty of the study”.
Responses: We are very grateful to you and your positive comments. We revised the novelty of the study in the revised manuscript.
Q12:Line 230-231 These points should be present in the introduction
Responses: We are very grateful to you and your positive comments. As suggested, we move these points into introduction in the revised manuscript.
Q13:Line 243-246 These points were already discussed, better to discuss your result and compare it with the similar findings from other published studies
Responses: We are very grateful to you and your positive comments. As suggested, we revised in the manuscript.
Q14:Line 253-257 not convinced with the written part, it should be written professionally, lastly, they directly suggested that "in the current study, please make a constant balance between the lines and provided information
Responses: We are very grateful to you and your positive comments. As suggested, we re-write and ask Wallace Academic Editing for language service, please find the language service certificate.
Q15:The journal style should be used by MDPI author guidelines. Abbreviated. please check all the references once again
Responses: Thanks very much for your great comments and suggestions. As suggested, we revised and checked all the references format according to the latest format requirements from author guideline.

Round 2
Reviewer 2 Report
None
Reviewer 3 Report
The authors Fang Shu-zhen, Yang Wen-ming, Zhang Kang-yi and Chuanyi Peng have revised the manuscript "Gandouling Tablets Mitigate the Copper-Overloaded-Induced Heart Injury in Rat" according to my comments. I accept the manuscript in its present form
Reviewer 4 Report
All comments has been addressed. Thank you